# Modeling Adsorption of CO_2_ in Rutile Metallic Oxide Surfaces: Implications in CO_2_ Catalysis

**DOI:** 10.3390/molecules28041776

**Published:** 2023-02-13

**Authors:** Rogelio Chávez-Rocha, Itzel Mercado-Sánchez, Ismael Vargas-Rodriguez, Joseelyne Hernández-Lima, Adán Bazán-Jiménez, Juvencio Robles, Marco A. García-Revilla

**Affiliations:** 1Chemistry Department, Natural and Exact Sciences Division, University of Guanajuato, Noria Alta S/N, Guanajuato 36050, Mexico; 2Pharmacy Department, Natural and Exact Sciences Division, University of Guanajuato, Noria Alta S/N, Guanajuato 36050, Mexico

**Keywords:** metallic oxide, CO_2_ adsorption, environmental remediation, DFT calculations

## Abstract

CO_2_ is the most abundant greenhouse gas, and for this reason, it is the main target for finding solutions to climatic change. A strategy of environmental remediation is the transformation of CO_2_ to an aggregated value product to generate a carbon-neutral cycle. CO_2_ reduction is a great challenge because of the large C=O dissociation energy, ~179 kcal/mol. Heterogeneous photocatalysis is a strategy to address this issue, where the adsorption process is the fundamental step. The focus of this work is the role of adsorption in CO_2_ reduction by means of modeling the CO_2_ adsorption in rutile metallic oxides (TiO_2_, GeO_2_, SnO_2,_ IrO_2_ and PbO_2_) using Density Functional Theory (DFT) and periodic DFT methods. The comparison of adsorption on different metal oxides forming the same type of crystal structure allowed us to observe the influence of the metal in the adsorption process. In the same way, we performed a comparison of the adsorption capability between two different surface planes, (001) and (110). Two CO_2_ configurations were observed, linear and folded: the folded conformations were observed in TiO_2_, GeO_2_ and SnO_2_, while the linear conformations were present in IrO_2_ and PbO_2_. The largest adsorption efficiency was displayed by the (001) surface planes. The CO_2_ linear and folded configurations were related to the interaction of the oxygen on the metallic surface with the adsorbate carbon, and the linear conformations were associated with the physisorption and folded configurations with chemisorption. TiO_2_ was the material with the best performance for CO_2_ interactions during the adsorption.

## 1. Introduction

CO_2_ is the most relevant greenhouse gas; the climatic change caused by high concentrations of this gas is related directly to human activity [1,2,3], and the impact of such climatic change is observed in the global economy and social wealth [4]. Several technological solutions were developed to reduce atmospheric emissions of CO_2_ [5,6,7], for instance, the use of emission gases as raw materials for low contaminant technological processes. The transformation of CO_2_ in aggregate-value products generates a carbon neutral cycle, which artificially reduces atmospheric emissions and relieves pressure on the traditional industry for energy production [8], considering the renewable production of combustibles. Nevertheless, a fundamental problem must be considered: CO_2_ is a highly stable molecule due to its linear geometry and efficient reduction potential, −1.9 V [9]. In addition, the C=O bond holds a dissociation energy of ~179 kcal/mol, which is around twice the C-H and C-C dissociation energies displaying values of 430 and 335 kJ/mol, respectively [10]. The reduction of CO_2_ is nonselective; depending on the reaction media and catalyzer, several chemical species are formed (CO, CH_4_, CH_3_OH, HCOOH, C_2_H_6_, C_2_H_5_OH) [11,12,13]. There are homogeneous and heterogeneous catalytic processes for CO_2_ reduction and activation [14,15,16,17]; nevertheless, heterogeneous catalysis has been proved to be more efficient and “clean” to obtain aggregate-value chemical species [18,19].

The synthesis of methane and methanol from CO_2_ and water has attracted great attention from the clean energy industry because of the enormous potential for the sustainable production of fuels using solar energy [20,21]. CO_2_ reduction mechanisms have been extensively studied in anatase and the rutile phases of TiO_2_ [22,23,24], and those results indicated that reduction occurs with multiple intermediates, suggesting that the disturbed geometry of CO_2_ during adsorption is the fundamental step for the reduction process. Another crucial factor is the efficiency of the catalyst, and relevant factors to consider are the charge separation displayed by the semiconductor [22,23] and the presence of water molecules during the process. The adsorption of water on the catalysts’ surface causes H_2_O dissociation [12,24,25,26], and such a process generates reactive species that are relevant in CO_2_ catalysis [26].

There are reports of semiconductor metal oxides capable of reducing CO_2_ in the presence of irradiation [24,27,28,29,30,31] despite the high reduction potential in the area that the reaction takes place. This indicates that the interaction between the CO_2_ and the semiconductor surface decreases the activation energy. Several experiments reported the detection of adsorbed CO_2_, which displayed modified structures [32,33,34,35,36]. DFT studies have analyzed the interactions between CO_2_ and different photocatalysts to study the role of adsorption on CO_2_ reduction [23,37,38,39,40,41,42,43,44,45,46].

Effective adsorption and activation are key steps for CO_2_ photoreduction [47], and adsorption models have been reported in the literature from kinetic and thermodynamic perspectives [48]. Therefore, the adsorption and subsequent activation of CO_2_ on the surfaces of heterogeneous catalysts are both crucial for the reduction process and for the suppression of the Hydrogen Evolution Reaction (HER). Under this perspective, the interactions of the CO_2_ molecule with the surface produce the formation of partially charged species (CO_2_^δ−^). Besides the linearity of gaseous carbon dioxide and the absence of a permanent dipole momentum, each oxygen atom has free electronic pairs that attack metal centers behaving as Lewis’s acids; therefore, the carbon atom is suitable to be attacked by a nucleophilic moiety. Considering these characteristics, CO_2_ has several coordination points, and this provides the possibility of having multiple geometries associated with the adsorption of CO_2_ on the surfaces of heterogeneous catalysts [37]. The morphology of the surface is relevant for the catalytic performance, and in addition, the adsorbed CO_2_ geometries are determined by the surface–adsorbate interactions, and the main anchor points of CO_2_ are the oxygen atoms. The preference of certain crystalline faces to adsorb CO_2_ has been observed, and this is favored for those surfaces with a large number of exposed metal atoms [38,39].

DFT studies showed that the coordination geometry of the surface controls the arrangement of the active sites, usually displaying cation behavior [40,41,42,43]. However, it is evident that due to the CO_2_ linearity, the possible adsorbed geometries are limited. Two adsorption configurations have been reported for CO_2_: linear and bent. The former occurs selectively on surfaces of certain metal oxides such as TiO_2_, CeO_2_, ZnO, SnO_2_ and CuO [41,42], while the latter is more common since it occurs on other metallic surfaces such as Au and Cu [41]. There is a third less common possibility in which a dissociation is carried out to M-CO and O-M during the CO_2_ adsorption, and the latter has been studied in detail using DFT calculations in surface models of Fe-Ni bimetallic catalysts [44].

Photochemical activation requires an electron transfer from the surface to the adsorbed CO_2_ molecule. The charge transfer from the metal to the CO_2_ is related to the way in which the adsorbate is coordinated, and this is the fundamental step of photoreduction. Ab initio calculations showed that bidentate CO_2_ adsorption on TiO_2_-anatase formed a hybrid LUMO between the CO_2_ orbitals and titanium *3d* orbitals. Such LUMO showed a decrease in energy compared with the LUMO of nonadsorbed CO_2_, which suggests that the adsorption favored the electronic transfer by the *3d* titanium orbitals to CO_2_ [43]. This ability to donate charge has also been studied and detected in TiO_2_ brookite models [45].

Surface defects are among the most reactive sites, and these sites experience electron density accumulation and change the surface reactivity properties. It has been proposed that oxygen vacancies play a vital role in several surface reactions [49], and those vacancies are particularly relevant in the adsorption and activation of CO_2_. Surface defects can promote higher-energy species during CO_2_ adsorption compared with perfect surfaces, although the defects are able to stabilize anionic CO_2_ [43]. Ab initio modeling has highlighted the importance of crystal defects for the description of adsorption and surface–adsorbate charge transfer for TiO_2_ models (brookite and anatase). Even so, it must be considered that perfect surfaces display similar adsorption properties to those with vacancies, at least regarding CO_2_ adsorption [23,45,46].

In this work, DFT studies were carried out on CO_2_ adsorption models on rutile-type structures from diverse metal oxides. We used one type of polymorph (rutile) to evince the cation’s role in adsorption, and we investigated the cation effect by comparing the adsorbed conformations and the cation effect in the adsorption energy. Additionally, we tested the adsorption in two different surfaces generated from two different lattice planes (001) and (110) to find the plane with the largest adsorption capability. Furthermore, we modeled the dissociative adsorption of water on these surfaces to analyze the competition for active sites with CO_2_, and finally, the formation of catalytic reactive species is proposed.

## 2. Methodology

We designed finite models of different metal oxide (MO) surfaces with rutile-like crystal structures. All of them were based on the structure reported in the crystallographic data banks followed by a geometry optimization; then, surface plane modeling was performed based on the Miller indices (001) and (110). The modeled MO surfaces were TiO_2,_ GeO_2_, SnO_2_, PbO_2_ and IrO_2_. In the case of TiO_2_, the anatase surfaces were also modeled based on the (001) and (110) Miller indices. Molecules of CO_2_ and H_2_O (in individual systems) that were previously optimized were added to the models. Subsequently, the geometry optimization of the entire system was carried out, and the lowest energy configuration was obtained. Modeling was performed using the ADF (Amsterdam Density Functional) computational package [50,51,52] and its supplement for periodic systems BAND [53,54,55,56] in version adf2014.01. The MO optimizations and adsorption modeling were performed using a TZP basis set with the PBEsol [57] functional with the D3 Grimme dispersion correction [58], which is a modification of the PBE functional [59] designed to obtain an improvement in the equilibrium properties of solids in bulk and surface systems [60]. The PBE functional was successfully used to study CO2 adsorption in metal oxide systems [40,41,42,43]. The adsorbates were previously optimized with a B3LYP/TZP level [61,62]. The ZORA pseudopotential [63,64,65,66] was used for relativistic corrections on the metals. The atomic charges were computed using the Hirshfeld scheme [67]. For details, see the Appendix A.

To build the surface of the finite models, we used the bulk structure of the metal oxide as the starting point and the crystallographic parameters available in the BAND software databank. The bulk structure was optimized using BAND software with the PBEsol-D3/TZP level of theory. Finally, the surface planes were obtained from Miller’s indices (001) and (110) using BAND to build the slabs. The adsorbates were individually optimized using the B3LYP/TZP level of theory. Afterwards, an optimized adsorbate slab was generated and then used to perform optimizations over the optimized solid bulk surface slab, and the initial distance between the adsorbate molecules and metallic center surfaces was 5 Å. Different initial adsorbate orientations were tested which resulted in similar optimized geometries, and the conformations with the lowest energy were used to perform the analysis. For details, see the Appendix A.

## 3. Results

### 3.1. CO_2_ Adsorption on TiO_2_ Surfaces

TiO_2_ belongs to the most studied substances related to the catalytic photoreduction of greenhouse gases due to its excellent photocatalytic properties. For this reason, we modeled the adsorption of CO_2_ on rutile and anatase phases, which are the most relevant phases for adsorption purposes [14]. Our results showed that both phases displayed an attractive interaction with CO_2_, and Figure 1 shows the optimized geometries for both studied phases for the rutile (001) and (110) planes and for the anatase (001) and (101) planes. The adsorbate geometry was bent in the rutile; meanwhile, in the anatase, the adsorbate geometry maintained its linearity. During the optimization, CO_2_ displayed a perpendicular orientation to the metallic surface; for this reason, an adsorbate oxygen atom (*O_A_*) approached directly to a surface titanium atom (*Ti*). At this stage of the optimization, the adsorption process differed for each phase. In the case of rutile, a second interaction appeared between the carbon and a surface oxygen (*O_S_*), and therefore the CO_2_ was bent; meanwhile, for the anatase, such an interaction and bending was not observed, and such results were the same for any plane (see Figure 2). The observed CO_2_ bent geometry in rutile is indicative of the capability to generate reactive species of CO_2_ during adsorption on the exposed surfaces.

The differences in the CO_2_ adsorbed structures, depending on the TiO_2_ phases, suggest that the capability of the surface to interact with the adsorbate is related to the phase topology. For the case of the (110) rutile plane, there was an oxygen out of the surface, which was the responsible of the interaction C--*O_S_*, denoted by the yellow oval on top of Figure 3. Additionally, rutile (001) displayed cavities between the connected layers and the alternation of the metallic atoms’ positions; consequently, there were regions with oxygens connecting layers that were capable of performing the C--*O_S_* interaction, as depicted by the blue oval on top of Figure 3. Nevertheless, the anatase (101) and (001) planes were flat surfaces without relevant cavities or prominences, and this situation reduced the exposition of the *O_S_* to have an interaction with the CO_2_ carbon.

### 3.2. CO_2_ Adsorption on SnO_2_, GeO_2_, PbO_2_ and IrO_2_ Rutile-Type Surfaces

The effect of the metallic cation on rutile-type surfaces on CO_2_ adsorption is a relevant issue to consider, and for this reason, we modeled the adsorption of CO_2_ on GeO_2_, SnO_2_, IrO_2_ and PbO_2_ rutile-type surfaces; the result of the optimization of CO_2_ over these surfaces is displayed in Figure 4. As in the case of TiO_2_, an attractive adsorbate–surface interaction was observed, displaying bended CO_2_ for the case of the GeO_2_ and SnO_2_ surfaces, and linear geometry for IrO_2_ and PbO_2_. Such behavior is apparently related to the weight of the metallic cation, as the surfaces with lighter metallic cations (GeO_2_and SnO_2_) displayed bended CO_2_ geometries and surfaces with heavier metallic cations (IrO_2_ and PbO_2_) displayed linear CO_2_ geometries. So, additionally to the topology of the metallic oxide surface, the size of the metallic cation must be considered for CO_2_ catalytic purposes. Larger sizes of metallic atoms (*M1)* sterically block accessibility to *O_S_* and neglect the possibility of a C--*O_S_* effective interaction, which is the critic step of the effective CO_2_ adsorption for catalytic purposes.

The adsorption energy is relevant when classifying a process as chemisorption or physisorption because weak intermolecular forces (van der Waals) are related with physisorption; meanwhile, chemisorption involves a structural transformation and formation (or elimination) of chemical bonds. Our criterion for the classification of adsorption is combining the energetic and structural values, whereby the largest adsorption energies with relevant structural changes in adsorbates are related to chemisorption; meanwhile, the lowest adsorption energies with an absence of structural changes are related to physisorption. The calculated values for the adsorption energies per CO_2_ molecule are displayed in Table 1. For the case of the (001) plane, a large interaction energy was displayed for the bended CO_2_ structures on TiO_2_, GeO_2_ and SnO_2_, followed by a less energetic adsorption for the linear CO_2_ structures on IrO_2_ and PbO_2_. A similar behavior was displayed for the (110) TiO_2_ and GeO_2_ planes, displaying the largest adsorption energies for the bended CO_2_ structures. However, the SnO_2_ and PbO_2_ showed similar adsorption energies displaying different CO_2_ adsorbed geometries, bended for SnO_2_ and linear for PbO_2_. The values of adsorption on the (001) planes were larger than those displayed by the (110) planes in general, except for the PbO_2_. In Table 2, some geometric parameters are displayed for the different adsorption geometries. It is evident that the distances *M1-O_A_* and *C-O_S_* were larger for the (110) planes in general, except for PbO_2_ and TiO_2_. A larger interaction distance was related to a weaker interaction, with the consequence of a lower interaction energy and less bending effect for the CO_2_ geometry. Therefore, the displayed adsorption energy for the exceptional case of PbO_2_ was related to the M1*-O_A_* distance, which was smaller for the (110) plane, which caused a larger adsorption energy.

The graph in Figure 5 shows the values of the adsorption energy against the bond distance M1--O_A_. On the left side of the graph, the presence of lighter metal oxides is observed, which shows that the shorter the bond distance, the greater the adsorption energy. This is consistent with the nature of this interaction, which is a Lewis acid–base reaction. Similarly, Figure 6 shows the adsorption energy graph against the change in angle of CO_2_. Again, there was a correspondence between the highest adsorption energies with the highest degree of modification in the structure of CO_2_. This suggests that both the first (M1--O_A_) and the second interactions (C_1_--O_S_) were the largest contributors to the adsorption energy. Our results suggest that chemisorption was observed for both planes of TiO_2_, GeO_2_ and SnO_2_, while physisorption was observed for both planes of IrO_2_ and for PbO_2_. Physisorption was related to the linear CO_2_ adsorbed geometries in general, and the bended CO_2_ adsorbed geometries were related to chemisorption, which is the expected behavior for a material with CO_2_ catalytic properties.

The strength of the adsorption of a molecule depends on the surface energy, which can be defined as the increase in the energy of the crystalline structure because of the generation of a surface, whereby an asymmetric environment is therefore produced. The bonding energies of the metallic oxides were calculated and are shown in Table 3, and such energies can be considered the relative surface energies because all the calculated metallic oxides held the same molecular structure and shared the number and type of dangling bonds. The results in Table 3 indicate that the (110) surfaces held larger negative energies that the (001) ones, which agree with the previous discussion on the chemisorption/physisorption properties of such planes. There was a trend observed by the elements of the 14(IVA) group whereby the binding energy increased with the period. Nonetheless, the transition metals did not follow the trend; the Ir surface should have displayed a larger binding energy than Ge and Sn surfaces because it is a heavier atom, but the Ir surface displayed lower binding energies than the Ge and Sn ones. This behavior was rationalized from the coordination sphere perspective. For the case of rutile, each cation was coordinated with six oxygen atoms and an octahedral geometry was obtained, isolating this six-coordinated system from the crystal. The combination of the ***s***, ***p*** and ***d*** orbitals generated ***d^2^sp^3^*** orbitals for the transition metals and ***sp^3^d^2^*** orbitals for the IVA group elements; therefore, the different electronic configurations for Ti and Ir displayed the same crystal arrangement. Additionally, for the case of the IVA group, the ***d*** orbitals participating in hybridization displayed the same ***n*** quantum number as the ***s*** and ***p*** orbitals. Meanwhile, the transition metal ***d*** orbitals were in a lower ***n-1*** level than the ***s*** and ***p*** orbitals; for this reason, the generated hybrid orbitals of the IVA group were higher in energy than those at an ***n-1*** level, and therefore, the IVA group surfaces displayed higher binding energies. The surface energy is a useful descriptor of the adsorption capability of a material; nevertheless, it is evident that the characterization of the interaction between the metallic cation and the adsorbed molecule is a relevant issue to consider.

### 3.3. Charge Redistribution and Effects over CO_2_

The redistribution of the charge density can be approximated by the analysis of the Hirshfeld atomic charges for the adsorption process. The change in atomic charges (AC) can be used as a measure of the charge transference during the adsorption process. In Figure 6 and Figure 7, the changes in charges for planes (001) and (110) are listed, as well as the metallic center (M1), the neighbored metallic atom (M2), the surface oxygen (O_S_) and the adsorbate oxygens (O_A_) and (O_B_). In Figure 8 we show a model of the charge redistribution based on the data in Figure 6 and Figure 7. Each CO_2_ adsorption conformation, bent or linear, displayed different charge redistribution mechanisms. In the case of the CO_2_ bent geometries, there were three regions where the change was larger than in the rest of the surface: the metallic cation (M1), responsible of the first interaction with adsorbate; the surface oxygen (O_S_), which bent the adsorbate through a second interaction with the adsorbate; and the CO_2_ carbon atom (C_1_), because of such a second interaction. In the case of linear geometries, the relevant charge transference was observed between O_A_ and the metallic cation M1.

For the CO_2_ bent geometries on plane (001), the initial interaction generated an electron density gain for M1, followed by a charge transference to the neighbored surface oxygens, labeled as O_3_ and O_4_ in Figure 9. Then, the attack of Os on the CO_2_ carbon generated a charge transference from O_S_ to C_1_. For the case of TiO_2_, the large charge gain was due to the largest CO_2_ deformation, which produced a large charge delocalization of ¶ electrons of CO_2_. Regarding the CO_2_ bent geometries in plane (001), a similar situation was observed; a gain in the electronic charge of M1 followed by a charge transference to O_3_ and O_4_, and then Os attacked the CO_2_ carbon. However, in this case, a large charge donation of the metallic cation M_2_ to O_S_ was observed. Along the series, the charge gaining of atom M1 was diminished and the transference to the adsorbate carbon was increased. For CO_2_ linear geometries in planes (001), the relevant charge transference was observed between the adsorbate oxygen (O_A_) and the metallic cation (M1). Meanwhile, for the linear geometries in the (110) plane, the charge transference of IrO_2_ to M1 was negligible, followed by a small charge change in the O_A_. Nevertheless, for PbO_2_, a significant charge gain was displayed for M1 obtained from O_A_. This analysis showed the relevance of the study on AC redistribution along the adsorption process.

The relevance of the adsorption in catalytic processes is to diminish the activation energy to reduce the CO_2_ molecule. To quantify how bent CO_2_ structures are less stable than linear ones, we performed calculation of single points of isolated molecules from the geometries of the generated species along the adsorption process. In addition, relative energies were calculated using the formula $\Delta =E_{linear}-E_{adsorbed}$, where $E_{linear}$ is the energy of the linear-isolated CO_2_ structure and $E_{adsorbed}$ is the energy of the isolated CO_2_ molecule, using the adsorbed geometry, Table 4. It is evident that the energies of the bent structures (from *TiO_2_, GeO_2_* and *SnO_2_* oxides) were larger than the linear species (from *IrO_2_* and *PbO_2_* oxides). Additionally, the relative energies shown in Table 4 displayed a similar behavior to the adsorption energies, whereby the species generated by the (001) plane were less stable than the (110) plane ones in general. TiO_2_ displayed the lowest difference in the relative energy between planes. There was a significant difference between the planes for the case of *GeO_2_* and *SnO_2_*, showing values of 37 and 25 kcal/mol, respectively. The *IrO_2_* and *PbO_2_* oxides displayed significant small values of relative energy; therefore, the adsorbed species on these surfaces displayed a large degree of stability. The electron affinity (EA) can be used to approximate the reduction potential using empirical parameters [68,69,70]. For this reason, we calculated EA using the expression $EA=E_{anion}-E_{neutral}$, where $E_{anion}$ is the energy of the anionic molecule and $E_{neutral}$ is the energy of the neutral molecule. We used the geometry of the adsorbed species and the geometry of the linear CO_2_ molecule as the reference. A comparison of the *EA* with the relative energies (D) and with the O-C-O angles showed that the large D was related to the large EA and had smaller O-C-O angles, which suggests the correlation between CO_2_ bending and the surface reduction capability.

### 3.4. Water Molecules Dissociation

The phenomenon of the adsorption of water molecules is important for several reasons. Water is a relevant substance in atmosphere phenomena as it plays a fundamental role in photocatalysis and electrochemistry on metal oxides and aqueous interfaces [71,72,73]. Several studies have shown that water molecules are completely dissociated on the surface of some metal oxides, generating hydroxylated surfaces [71,74,75,76,77]. TiO_2_ and SnO_2_ rutile are capable of dissociating H_2_O [32,74,78]. The (110) plane adsorbs H_2_O in a dissociative manner and the proton is attached to one of the bridge oxygens of the surface while the hydroxyl is attached to the metal center [79]. Water molecular adsorption is possible through hydrogen bond formation with OH groups; nevertheless, it makes it difficult to estimate the dissociation energy from experimental results. Gercher and Cox [78] used thermal desorption spectroscopy (TDS) and ultraviolet photoemission techniques to study water adsorption on surfaces. Their results showed that water is absorbed in a dissociative conformation on a perfect surface; this fact was attributed to a peak at 435 K in the desorption spectrum, and peaks at 200 and 300 K were also attributed to molecular water desorption.

We performed H_2_O adsorption processes over TiO_2_, GeO_2_ and SnO_2_ surfaces, and we did not consider IrO_2_ and PbO_2_ because these oxides display low adsorbate affinity. The adsorption models are displayed in Figure 10. The active site was the same as in the case of CO_2_ adsorption and similar processes occurred during H_2_O adsorption; there were two surface interactions that formed M1-OH and O_S_-H, and this observation agreed with the results reported by Lindan [79]. The surface dissociation energies of the water molecules dissociating into metal oxides are shown in Table 5; our values were larger than those reported in previous studies [71,79,80], in which the dissociation of H_2_O on SnO_2_ and TiO2 (001) displayed values around −40 kcal/mol. Table 5 shows that the process of H_2_O surface dissociation was preferred over the adsorption of CO_2_, considering that H-OH bond breaking took place. The study of the processes that occur after surface H_2_O dissociation is needed to consider what the probable reactive species for CO_2_ catalysis are. A relevant question must be addressed: is there a chance of liberation of H and OH as free radicals? Or do these species stay on the surface? To obtain a close connection with the experimental conditions of photocatalysis, adsorbates mixes must be modeled (H_2_O and CO_2_) on surfaces together with a characterization of vertical excitations, excited states relaxation and electromagnetic emissions. This study will be addressed in future work.

## 4. Conclusions

The results showed two adsorbate geometries for CO_2_ adsorption on rutile and anatase surfaces: one of them was a bended structure and the second one was linear. The observation of one of these geometries depended on the interaction between a surface oxygen and the carbon atom of adsorbate (C-Os) as this interaction was responsible for the CO_2_ bending. The main difference between the adsorption behavior of the TiO_2_ phases was the availability of the surface oxygens to interact with the adsorbate. Additionally, the surface topology was relevant to perform secondary interactions with adsorbates. The studied oxides (TiO_2_, GeO_2_, SnO_2_, IrO_2_ and PbO_2_) were capable of adsorbing CO_2_; nevertheless, the lighter oxides, TiO_2_, GeO_2_ and SnO_2_, bent CO_2_ during adsorption. Meanwhile, the heaviest oxides (IrO_2_ and PbO_2_) maintained a linear CO_2_ geometry. Chemisorption was observed for both planes of TiO_2_, GeO_2_ and SnO_2_, while physisorption was observed for both planes of IrO_2_ and PbO_2_. Therefore, the size of the metallic cation was relevant to the final adsorbed geometries. The planes (001) displayed the largest CO_2_ affinities compared with the (110) planes. Additionally, the bent CO_2_ geometries were related to chemisorption processes, lower CO_2_ stability and a large positive and therefore a positive reduction potential. Finally, it was found that the water molecules dissociated during the adsorption process on the TiO_2_, GeO_2_ and SnO_2_ surfaces, displaying the same active sites as in the case of CO_2_ adsorption, whereby the M1-O-H and O_S_-H species were observed in the adsorbed H_2_O. These reactive species with bent CO_2_ geometries suggested the generation of multiple secondary products that will be studied in future work.

## Figures and Tables

**Figure 1 molecules-28-01776-f001:**
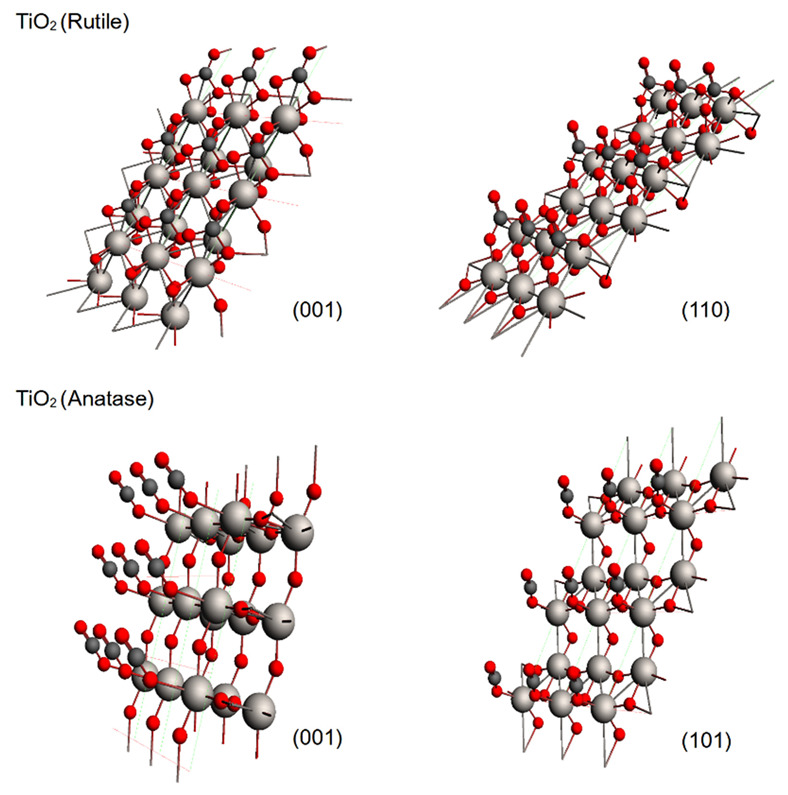
Optimized geometries for the adsorption models over the crystalline phases of TiO_2_: **top:** (001) and (110) rutile planes; **bottom**: (001) and (101) anatase planes. Atomic representations: titanium atoms as white spheres, oxygen atoms as red spheres and carbon atoms as dark gray spheres.

**Figure 2 molecules-28-01776-f002:**
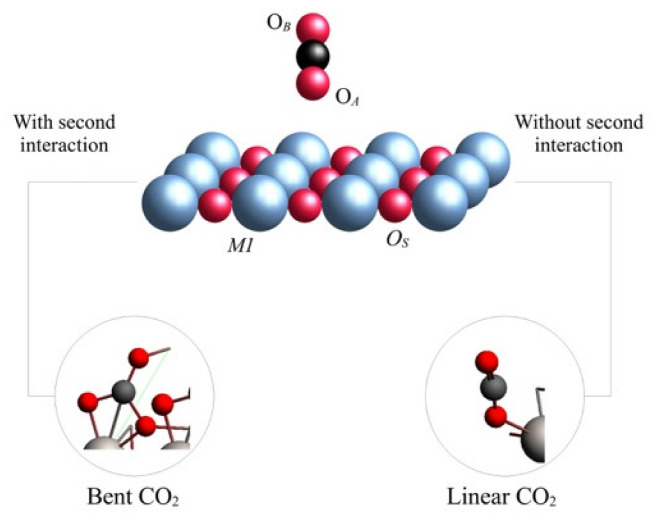
Representation of CO_2_ adsorption on TiO_2_ on rutile (**right**) and anatase (**left**). Atoms are labeled as follows: O_A_ is the oxygen atom of CO_2_ which displays the first interaction with the surface; O_B_ is the second oxygen atom of CO_2_; O_S_ is the metallic oxide oxygen; M1 is the metallic center.

**Figure 3 molecules-28-01776-f003:**
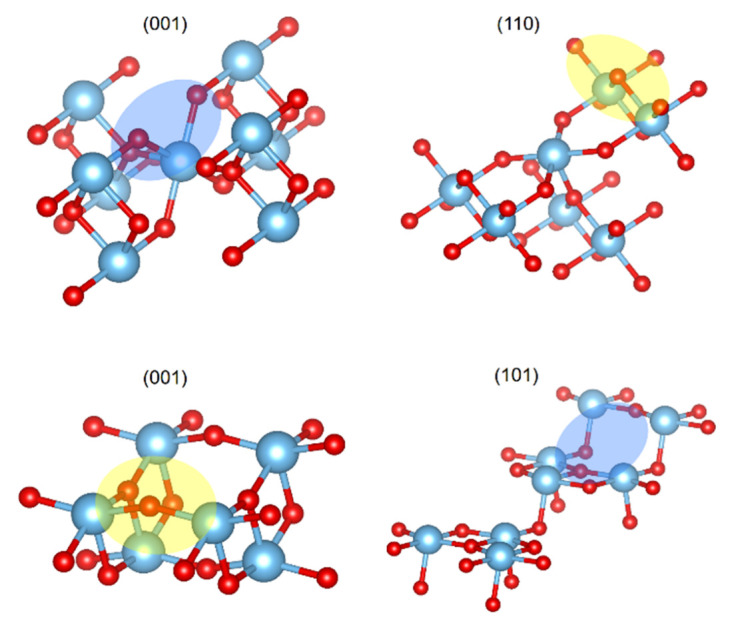
Representation of rutile (**top**) and anatase (**bottom**) studied planes. Regions top of the plane in yellow ovals and regions below of the plane in blue ovals. Titanium atoms are depicted as blue spheres and oxygen as red spheres.

**Figure 4 molecules-28-01776-f004:**
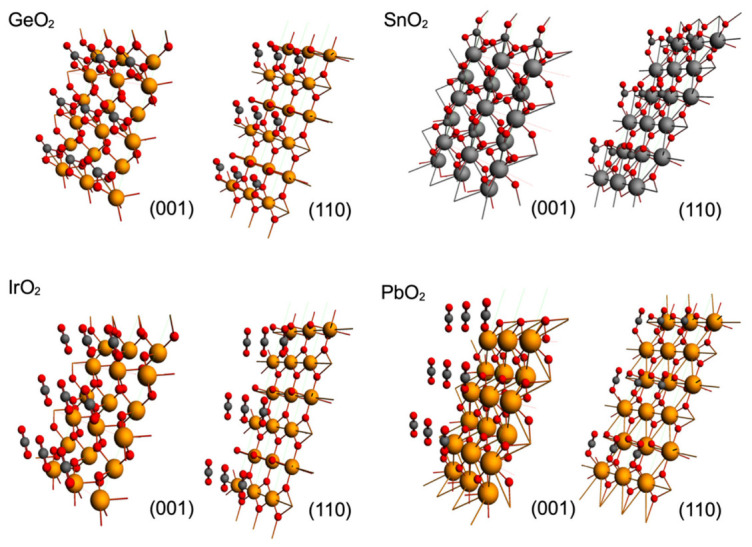
Optimized geometries for GeO_2_, SnO_2_, IrO_2_ and PbO_2_ rutile-type surfaces on (001) and (001) planes. Metallic cations are shown as large orange spheres, oxygen atoms as red spheres and carbons as dark grey spheres.

**Figure 5 molecules-28-01776-f005:**
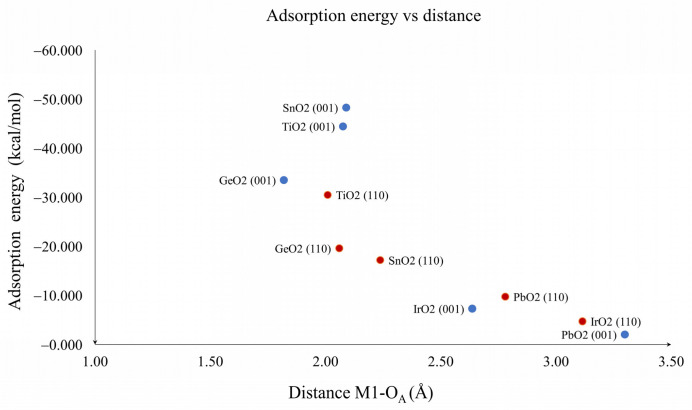
Relation between adsorption energy and M1-O_A_ distance for TiO_2_, GeO_2_, SnO_2_, IrO_2_ and PbO_2_. (001) planes are blue dots and (110) planes are red dots.

**Figure 6 molecules-28-01776-f006:**
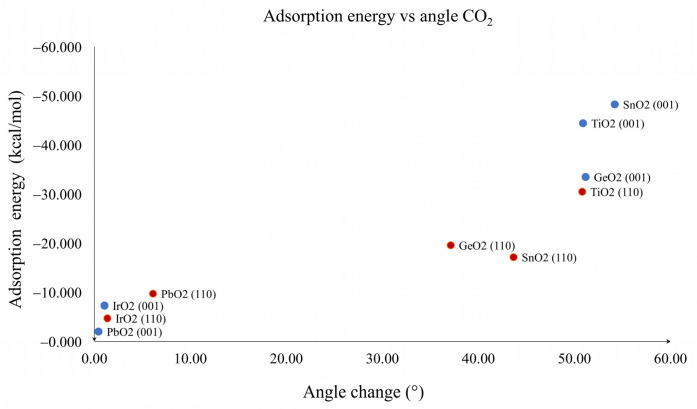
Relation between adsorption energy and CO_2_ angle for TiO_2_, GeO_2_, SnO_2_, IrO_2_ and PbO_2_. (001) planes are blue dots and (110) planes are red dots.

**Figure 7 molecules-28-01776-f007:**
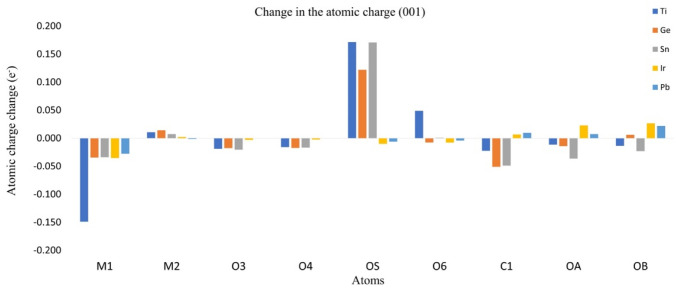
Atomic charge change for plane (001).

**Figure 8 molecules-28-01776-f008:**
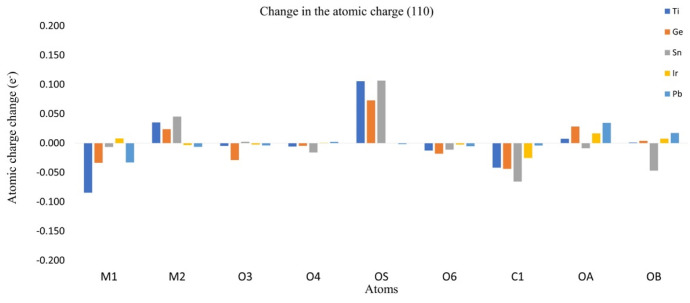
Atomic charge change for plane (110).

**Figure 9 molecules-28-01776-f009:**
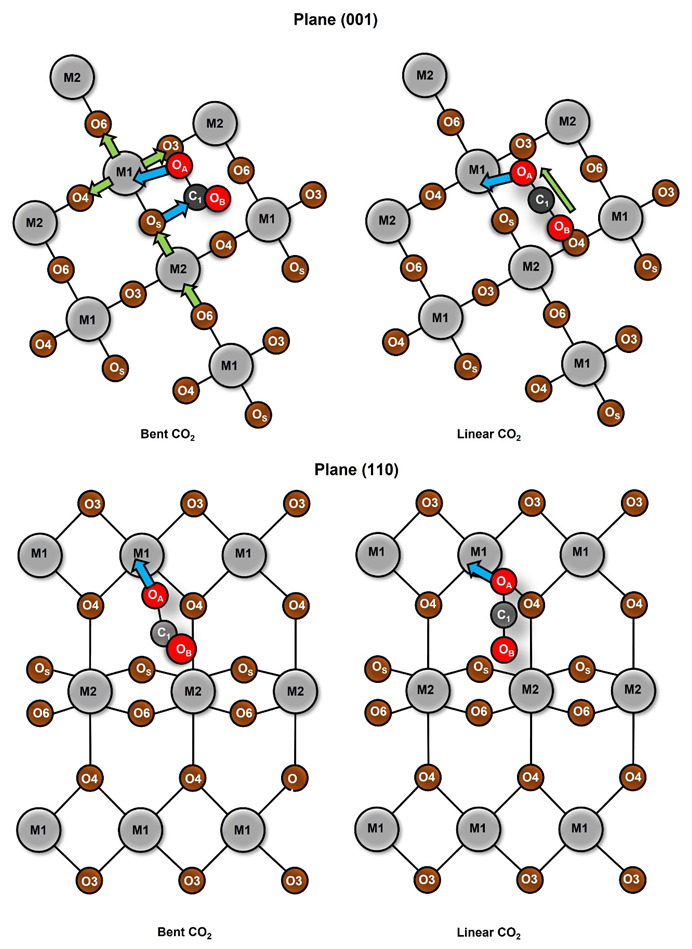
Mechanism of atomic charge redistribution for plane (001) **left** and (110) **right**.

**Figure 10 molecules-28-01776-f010:**
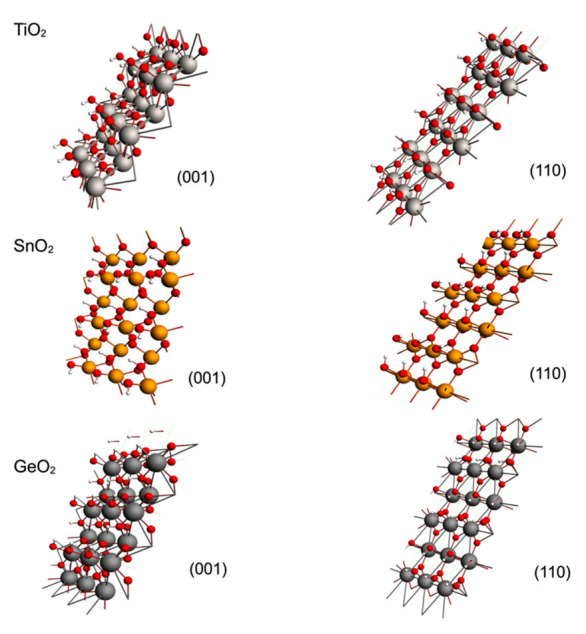
H_2_O adsorption–dissociation geometries for TiO_2_, GeO_2_ and SnO_2_ surfaces in planes (001) and (110). Oxygen are red spheres, metals are large gray spheres and hydrogen are small white spheres.

**Table 1 molecules-28-01776-t001:** Adsorption CO_2_ energies for TiO_2_, GeO_2_, SnO_2_, IrO_2_ and PbO_2_ on (001) and (110) planes.

Oxide	Plane (001) (kcal/mol)	Plane (110) (kcal/mol)
TiO_2_	−44.420	−30.450
GeO_2_	−33.447	−19.540
SnO_2_	−48.242	−17.113
IrO_2_	−7.249	−4.671
PbO_2_	−1.982	−9.663

**Table 2 molecules-28-01776-t002:** Distances and angles of the adsorption geometries for TiO_2_, GeO_2_, SnO_2_, IrO_2_ and PbO_2_.

Oxide	Angle O-C-O	Distance C-O_A_ (Å)	Distance C-O_B_ (Å)	Distance C-O_S_ (Å)	Distance M-O_A_ (Å)
TiO_2_ (001)	129.1	1.27	1.24	1.39	2.08
TiO_2_ (110)	129.2	1.31	1.19	1.60	2.01
GeO_2_ (001)	128.8	1.35	1.19	1.44	1.82
GeO_2_ (110)	142.9	1.25	1.17	1.94	2.06
SnO_2_ (001)	125.8	1.29	1.24	1.40	2.09
SnO_2_ (110)	136.3	1.27	1.19	1.74	2.24
IrO_2_ (001)	178.9	1.18	1.17	2.98	2.64
IrO_2_ (110)	178.6	1.17	1.17	2.89	3.12
PbO_2_ (001)	179.6	1.17	1.17	4.25	3.30
PbO_2_ (110)	173.9	1.18	1.17	2.69	2.78

**Table 3 molecules-28-01776-t003:** Bonding energies for (001) and (110) planes of TiO_2_, GeO_2_, SnO_2_, IrO_2_ and PbO_2_.

Oxides	E_B_ (001) kcal /mol	E_B_ (110) kcal /mol
TiO_2_	−182.42	−186.33
GeO_2_	−128.66	−139.57
SnO_2_	−121.73	−131.80
IrO_2_	−149.97	−162.54
PbO_2_	−106.52	−112.86

**Table 4 molecules-28-01776-t004:** Relative energy (D) of CO_2_ species generated during adsorption; electron affinities (EA) of adsorbed CO_2_ molecules. The linear CO_2_ molecule display EA of −4.19 eV.

Oxide	(001) D (kcal/mol)	(110) D (kcal/mol)	(001) EA (eV)	(110) EA (eV)
TiO_2_	57.433	59.433	1.28	1.20
GeO_2_	66.058	29.763	1.33	−0.16
SnO_2_	66.182	41.964	1.57	0.55
IrO_2_	0.596	0.607	−3.89	−3.88
PbO_2_	0.232	1.177	−4.00	−3.51

**Table 5 molecules-28-01776-t005:** H_2_O surface dissociation energies in TiO_2_, GeO_2_ and SnO_2_ oxides for (001) and (110) planes.

Oxide	E_D_ (001) (kcal/mol)	E_D_ (110) (kcal/mol)
TiO_2_	−31.508	−71.164
SnO_2_	−60.094	−56.541
GeO_2_	−62.578	−57.525

## Data Availability

Not applicable.

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
