# Peer review of "Modeling Adsorption of CO2 in Rutile Metallic Oxide Surfaces: Implications in CO2 Catalysis"

_molecules, 2023, doi:10.3390/molecules28041776_

Round 1

Reviewer 1 Report (Previous Reviewer 1)

In the revised manuscript, I found that the authors appropriately recalculated the CO2-adsorbed rutile-metallic oxide surfaces with a dispersion correction.

However, there is still a point that should be revised for the publication. The experimental results of H2O adsorption energies, which must be observed, should be compared to the calculated results in Table 5 in order to confirm the reliability of the calculation model.

Author Response

Point-by-point Answers to Reviewers

Dear reviewer,

Thank you for your observations, the quality of the manuscript has been greatly increased because of that.

Please find below a point-by-point answers to your observations.

In the revised manuscript, I found that the authors appropriately recalculated the CO2-adsorbed rutile-metallic oxide surfaces with a dispersion correction.

However, there is still a point that should be revised for the publication. The experimental results of H2O adsorption energies, which must be observed, should be compared to the calculated results in Table 5 in order to confirm the reliability of the calculation model.

Answer:

We have included the following text to show the available information related with this topic

“The phenomenon of adsorption of water molecules is important for several reasons. Water is a relevant substance in atmosphere phenomenon, it plays a fundamental role in photocatalysis and electrochemistry on metal oxides and aqueous interfaces [71-73]. Several studies have shown that water molecules are completely dissociated on the surface of some metal oxides, generating hydroxylated surfaces [71,74-77]. TiO2 and SnO2 rutile are capable of dissociating H2O [32,74,78]. The (110) plane adsorbs H2O in a dissociative manner, the proton is attached to one of the bridge oxygens of the surface, while the hydroxyl is attached to the metal center [79]. Water molecular adsorption is possible through hydrogen bond formation with OH groups, nevertheless, it makes difficult to estimate the dissociation energy from experimental results. Gercher and Cox [78] used thermal desorption spectroscopy (TDS) and ultraviolet photoemission techniques for the study of water adsorption on surfaces. Their results show that water adsorbs in a dissociative conformation on a perfect surface, attributing this fact to a peak at 435 K in the desorption spectrum, peaks at 200 and 300 K were also attributed to molecular water desorption.

We have performed H2O adsorption processes over TiO2, GeO2 and SnO2 surfaces, we have not considered IrO2 and PbO2 because these oxides display low adsorbate affinity. The adsorption models are displayed in Figure 9. The active site is the same as in the case of CO2 adsorption and similar processes occur during H2O adsorption, there are two surface interactions formed M1-OH and OS-H, this observation agrees with the reported by Lindan [79]. The surface dissociation energies of water molecules into metal oxides are shown in Table 5, our values are larger than those reported in previous studies [71,79,80], in which the dissociation of H2O on SnO2 and TiO2 (001) display values around -40 kcal/mol. The Table 5 show that the process of H2O surface dissociation is preferred over the adsorption of CO2, considering that H-OH bond breaking takes place.”

Best regards,

Marco Antonio García Revilla.

Reviewer 2 Report (Previous Reviewer 3)

Comments from Reviewer

Title: Modelling adsorption of CO2 in rutile-metallic oxide surfaces; implications in photocatalysis

I want to point out that I am reviewing the paper for the second time. After checking the manuscript, it turns out that the authors did not scan it for editing errors. The current form's presentation of methods and scientific results is satisfactory for publication in the Molecules journal. The minor drawbacks to be addressed can be specified as follows:
1.    A large number of minor errors/typos, eg. (i) Line 33. ".[4]” ---> “[4].” (ii) Line 43. “.[10]” ---> "[10]." (iii) line 93, TiO2,CeO2,ZnO – no spaces (iv) Line 108, "(TiO2) [48]- no dot and problem with ")". The reviewed article must be thoroughly checked before being published.
2.    Page 5. This page is not showing.
3.    The lack of pages with lines 181-258.
4.    Figs. 1-4??????
5.    Unfortunately, I don't see this page, but it seems to me that this comment is still valid: Fig. 1, figure captions. (i) speres or spheres? (ii) In my opinion, white grey large speres are white spheres. See also Line 407.
6.    Unfortunately, I don't see this page, but it seems to me that this comment is still valid: Fig. 2. Oa, Ob, Os??? Please explain.
7.    Unfortunately, I don't see this page, but it seems to me that this comment is still valid: Fig. 4, carbons as small black spheres. They are dark grey spheres.
8.    Unfortunately, I don't see this page, but it seems to me that this comment is still valid: Tab. 1. Have the authors tried to look for a correlation between adsorption CO2 energy and some parameters characterizing the adsorbent? I do not mean geometrical parameters (Fig. 5) but, for example, the charge, hardness/softness of the adsorption centres.
9.    Unfortunately, I don't see this page, but it seems to me that this comment is still valid:Tab. 2. 178 ---> 178.0, 180 ---> 180.0; see also 2.24 and 3.37.
10.    Line 335. Atomic charges ---> Atomic charges (CA).
11.    Tab. 4. (i) area  ---> As. (ii) Please, give units for As and charge density index. What about CA? [e]?
12.    Line 346 and Fig. 2, the metallic centre (M1), the surface oxygen (Os). This excuse comes up too late !!!!!
13.    Line 349. OA or Oa??? Check the manuscript.
14.    Figs. 6 and 7. (i) y-axis, AC or CA? (ii) x-axis. The description of this axis is illegible. Please enlarge the font.
15.    Line 370. Elinear? Eadsorbed?

Sincerely,
    The reviewer.

Author Response

Point-by-point Answers to Reviewers

Dear reviewer,

Thank you for your observations, the quality of the manuscript has been greatly increased because of that.

Please find below a point-by-point answers to your observations.

I want to point out that I am reviewing the paper for the second time. After checking the manuscript, it turns out that the authors did not scan it for editing errors. The current form's presentation of methods and scientific results is satisfactory for publication in the Molecules journal. The minor drawbacks to be addressed can be specified as follows:
1.    A large number of minor errors/typos, eg. (i) Line 33. ".[4]” ---> “[4].” (ii) Line 43. “.[10]” ---> "[10]." (iii) line 93, TiO2,CeO2,ZnO – no spaces (iv) Line 108, "(TiO2) [48]- no dot and problem with ")". The reviewed article must be thoroughly checked before being published.

Answer:

We are sorry about this, we have checked and corrected all typos and minor errors along the manuscript

  1. Page 5. This page is not showing.

Answer:

We have corrected the page numbers counting.

  1. The lack of pages with lines 181-258.

Answer:

We have corrected the document format to solve this issue.

  1. Figs. 1-4??????
    5.    Unfortunately, I don't see this page, but it seems to me that this comment is still valid: Fig. 1, figure captions. (i) speres or spheres? (ii) In my opinion, white grey large speres are white spheres. See also Line 407.

Answer:

To solve 4 and 5 observations, in the revised version of the manuscript we have used the recommended format for the caption in all figures and in the text along the entire manuscript.

  1. Unfortunately, I don't see this page, but it seems to me that this comment is still valid: Fig. 2. Oa, Ob, Os??? Please explain.

Answer:

We have provided a new figure to avoid ambiguities. In addition, please find before the figure in page 4 a detailed explanation of the used labels. It reads:

 “During the optimization, CO2 displays a perpendicular orientation to the metallic surface, for this reason, an adsorbate oxygen atom (OA) approaches directly to a surface titanium atom (Ti). At this stage of the optimization, the adsorption process differs from each phase. For the case of rutile, a second interaction appears between the carbon and a surface oxygen (OS), therefore the CO2 is bent, meanwhile, for anatase such interaction and bending is not observed, see Figure 2, such results are the same for any plane.”

In addition, the new caption of figure 2 reads:

Figure 2. Representation of CO2 adsorption on TiO2 on rutile (right) and anatase (left). Atom labels as follows: OA is the oxygen atom of CO2 which displays the first interaction with the surface; OB is the second oxygen atom of CO2; OS is the metallic oxide oxygen; M1 is the metallic center.”

  1. Unfortunately, I don't see this page, but it seems to me that this comment is still valid: Fig. 4, carbons as small black spheres. They are dark grey spheres.

Answer:

In the revised version of the manuscript, we have used the recommended format for the caption.

  1. Unfortunately, I don't see this page, but it seems to me that this comment is still valid: Tab. 1. Have the authors tried to look for a correlation between adsorption CO2 energy and some parameters characterizing the adsorbent? I do not mean geometrical parameters (Fig. 5) but, for example, the charge, hardness/softness of the adsorption centres.

Unfortunately, we haven’t found a robust correlation between any descriptor and the adsorption energy, we have tried to find a correlation, nevertheless, the observed trends display low correlation indices.  

  1. Unfortunately, I don't see this page, but it seems to me that this comment is still valid:Tab. 2. 178 ---> 178.0, 180 ---> 180.0; see also 2.24 and 3.37.

Answer:

In the revised version of the manuscript, we have used the recommended format for the significant digits.

  1. Line 335. Atomic charges ---> Atomic charges (CA).

Answer:

In the revised version of the manuscript, we have used “Atomic Charges (AC)”.

  1. Tab. 4. (i) area  ---> As. (ii) Please, give units for As and charge density index. What about CA? [e]?

Answer:

In the revised version of the manuscript, we have eliminated that section, and the density index no longer appear.

  1. Line 346 and Fig. 2, the metallic centre (M1), the surface oxygen (Os). This excuse comes up too late !!!!!

Answer:

We think that the reviewer observation is related with the charge transference mechanism, in the revised version of the manuscript a detailed discussion of this mechanism is placed. Please find a figure, Figure 9, and the following text in page 11.

“For CO2 bent geometries on planes (001) the initial interaction generates an electron density gain for M1, followed by a charge transference to the neighbored surface oxygens, labelled as O3 and O4 in Figure 9. Then the attack of Os to the CO2 carbon generates a charge transference from OS to C1. For the case of TiO2, the large charge gaining is due to the largest CO2 deformation, which produces a large charge delocalization of ¶ electrons of CO2. Regarding CO2 bent geometries in planes (001), a similar situation is observed; a gain in electronic charge of M1, followed by a charge transference to O3 and O4, then Os attacks the CO2 carbon, however, in this case a large charge donation of the metallic cation M2 to OS is observed. Along the series, the charge gaining of atom M1 is diminished and the transference to the adsorbate carbon is increased. For CO2 linear geometries in planes (001), the relevant charge transference is observed between the adsorbate oxygen (OA) and the metallic cation (M1). Meanwhile, for linear geometries in (110) planes the charge transference to M1 of IrO2 is negligible, followed by a small charge change of the OA, nevertheless, for PbO2 a significant charge gaining is displayed for M1 obtained from OA. This analysis shows the relevance of the study of the AC redistribution along the adsorption process.”

  1. Line 349. OA or Oa??? Check the manuscript.

Answer:

We have used OA in the revised version of the manuscript.

  1. Figs. 6 and 7. (i) y-axis, AC or CA? (ii) x-axis. The description of this axis is illegible. Please enlarge the font.

Answer:

We have used the label “atomic charge change” in the cited figures, Figures 7 and 8 in the revised version of the manuscript. In addition, we have enlarged the fonts of y-axis and x-axis. In addition, we have enlarged the fonts of figures 5 and 6.

  1. Line 370. Elinear? Eadsorbed?

    Answer:

We have defined such quantities in the revised version of the manuscript, please find the following thext in page 13.

“In addition, relative energies were calculated using the formula , where  is the energy of the linear-isolated CO2 structure and  is the energy of the isolated COmolecule, using the adsorbed geometry, Table 4.”

Best regards,

Marco Antonio García Revilla.

Round 2

Reviewer 2 Report (Previous Reviewer 3)

Comments from Reviewer

Title: Modelling adsorption of CO2 in rutile-metallic oxide surfaces; implications in photocatalysis

Congratulations on a great job. The author has made a substantial improvement for this article. The manuscript can be accepted for publishment in the present form. However, the minor drawbacks to be addressed can be specified as follows:
1.    Supplementary, Fig. 1. (i) Geometry ---> geometry (ii) Models ---> models.
2.    Fig. 2. This figure is invisible.
3.    Figs. 7 and 8, figure captions. Atomic Charge ---> Atomic charge.

Sincerely,
   The reviewer.

Author Response

Point-by-point Answers to Reviewers

Congratulations on a great job. The author has made a substantial improvement for this article. The manuscript can be accepted for publishment in the present form.

Dear reviewer,

Thank you for your kind answer, your observations were extremely helpful to improve the quality of our manuscript.

Please find below a point-by-point answers to your observations.

However, the minor drawbacks to be addressed can be specified as follows:
1.    Supplementary, Fig. 1. (i) Geometry ---> geometry (ii) Models ---> models.

Answer: We have changed to lowercase the cited words in the supplementary information file.

  1. Fig. 2. This figure is invisible.

Answer: we have increased the size of Figure 2.

  1. Figs. 7 and 8, figure captions. Atomic Charge ---> Atomic charge.

Answer: we have changed the  caption of Figures 7 and 8.

Best regards,

Marco Antonio García Revilla.

This manuscript is a resubmission of an earlier submission. The following is a list of the peer review reports and author responses from that submission.

Round 1

Reviewer 1 Report

In this manuscript, the authors calculated the adsorption energies and other several properties for the CO2 adsorption on some rutile-metallic oxide surfaces. This study intends to explore the photocatalytic reactions of these surfaces. However, it is unclear how this study contributes to the investigations of the photocatalytic reactions. Besides, there are many serious problems in the calculations and the discussions of the results. I, therefore, recommend that this manuscript should be rejected. Followings are the points that have to be revised:

1. The authors should first show the photocatalysis mechanism of CO2 decomposition on these surfaces to explain the significance of the CO2 adsorption calculations of these surfaces. It is also meaningful for this study to show experimental evidences that the photocatalytic reaction rares depend on the adsorption rates.

2. This study uses PBEsol functional in the calculations. However, this functional contains no van der Waals effects, though the authors discuss the physisorptions of CO2. In addition, the criteria is not clarified for determining if the adsorptions are physisorptions or chemisorptions.

3. No experimental results is compared to support the calculation method. I suspect that this functional is too poor to quantitatively discuss the adsorption energies. This study needs some evidences for disproving this suspect.

4. Many typos are included: e.g, "the roll or adsorption", "the most unfamous greenhouse effect", and "larger negative energies that .."

Reviewer 2 Report

In this paper, the authors use density functional theory to explore how the adsorption of CO2 onto metal-oxide surfaces is affected by the identity of the metal as well as the plane of the surface. Through analysis of the adsorption energies, geometric changes, and partial charges, the authors develop trends to predict which metal oxide catalysts will most activate CO2 for subsequent reaction. The photocatlytic activation of CO2 is an important area and I believe this work has the potential to be publishable. However, I cannot recommend publication at this time until major revisions are performed.

Major Issues

1) As the authors admit in the introduction, a large number of previous experimental and computational studies have been performed on the adsorption of CO2 onto metal oxide surfaces. The authors need to clearly state what new science is being performed in this study. How does this study contribute to the field?

2) The description of the computational methods are very incomplete. The authors need to clarify: a) Details of their model for the metal oxide surface. How large of a surface is explicitly included in the calculation? How many layers? Some of this information appears to be in the supporting information but it also needs to be stated in the main text as well. b) Were periodic boundary conditions used? The methods section begins by stating that the authors used “finite models of surfaces of different metal oxides” but later talked about using “ADF and its supplement for periodic systems BAND”. c) In the text of the paper the authors state that “Subsequently, the geometry optimization of the entire system was carried out until the configuration of the lowest possible energy was obtained.” In the supporting information the authors state “To simulate the behavior of the solid, the geometries of the surfaces were fixed, leaving the adsorbate molecules free.” These two statements are contradictory. The authors need to clearly and accurately describe what they did. d) In the supporting information, the authors state that their surfaces were based on a previously optimized bulk model. If this model was published, then the authors need to cite it. If not, then the authors need to provide additional information about this in their work. e) What CO2 coverage was used? Based on the figures later in the paper, it appears as if the surface was completely covered in CO2.

3) It appears that the authors did not utilize a dispersion corrected functional. This seems troublesome given that the authors are looking at non-covalent interactions (i.e. adsorption). The authors need to comment on this. At minimum, I believe that the authors should perform some testing to verify that their results are not significantly affected by the inclusion of dispersion corrections.

4) Throughout the paper, the authors utilize a variety of different terminologies for bent CO2 (folded, crooked, etc.). The authors should consistently refer to these geometries as “bent” to be clear.

5) The text surrounding Table 4 as well as the Conclusions state that rR agrees with the trend in the adsorption energies (Table 1). This claim is simply not supported by the data. For example: a. TiO2 (001) and PbO2 (110) have the same rR despite having totally different adsorption geometries (bent versus linear) and dramatically different adsorption energies (-49.000 kcal/mol versus -14.491 kcal/mol). b. TiO2 (110) has a larger rR than TiO2 (001) despite having a weaker adsorption energy. Moreover, I plotted rR versus adsorption energy using the data in the paper and did not find a meaningful trend: -60 -50 -40 -30 -20 -10 0 0 0.005 0.01 0.015 0.02 0.025 0.03 0.035 0.04 0.045 (see attached file)

6) I have two questions/comments on section 3.4 Charge redistribution and effects over CO2: a. Figure 6 shows that for TiO2 (001), the dominate charge redistribution is a loss of electron density for OS, a gain in the electron density for C, and a gain in electron density for M1. The loss of electron density for OS and gain in electron density for C makes physical sense. However, the gain in electron density for M1 is hard for me to wrap my head around, especially as it is nearly twice that of the C. The authors should provide a mechanistic explanation for this as well as the fact that it is so much larger for TiO2 than for the other metals. b. Line 349 does not make sense as written. There is no evidence in the figures for electron transfer from OA to OS. Indeed, OS loses electron density in the adsorption process.

7) Throughout this paper, the authors appear to only consider complete coverage of the surface by either CO2 or H2O. This extreme situation does not seem to me to be relevant for catalysis. I strongly recommend that the authors consider how the energetics and partial charges are affected by changing the coverage on the surface. At a minimum, the authors need to discuss the extent to which their results connect to typical experimental conditions.

8) The paper will benefit from editing by a native English speaker.

Minor Issues

1) In descripting Table 1, the authors should explicitly state that the adsorption energies are given per CO2.

2) In Figure 3, I recommend that the authors add labels to indicate which row is rutile and which row is anatase.

3) In Figure 5, the authors color things based on the ratio of the energy to the M-Os distance. It is unclear to me why this ratio is physically meaningful? A more useful coloring, in my mind, would be based on the CO2 bond angle.

4) I recommend that the authors revise Figure 7 to use a y-axis label that better represents that the y-axis is the change in charge.

5) In addition to the analysis presented in Table 5, the authors may want to consider the reduction potential of CO2 to demonstrate that the adsorption to the surface will make CO2 easier to reduce.

Reviewer 3 Report

Comments from Reviewer
Title: Modelling adsorption of CO2 in rutile-metallic oxide surfaces; implications in photocatalysis
The current form's presentation of methods and scientific results is satisfactory for publication in the Molecules journal. The minor drawbacks to be addressed can be specified as follows:
1.    A large number of minor errors/typos, eg. (i) Line 33. “.[4]” ---> “[4].” (ii) Line 43. “.[10]” ---> “[10].” (iii) line 93, TiO2,CeO2,ZnO – no spaces (iv) Line 108, “(TiO2) [48]- no dot and problem with “)”. The reviewed article must be thoroughly checked before being published.
2.    Fig. 1, figure captions. (i) speres or spheres? (ii) In my opinion, white grey large speres are white spheres. See also Line 407.
3.    Fig. 2. Oa, Ob, Os??? Please explain.
4.    Fig. 4, carbons as small black spheres. They are dark grey spheres.
5.    Tab. 1. Have the authors tried to look for a correlation between adsorption CO2 energy and some parameters characterizing the adsorbent? I do not mean geometrical parameters (Fig. 5) but, for example, the charge, hardness/softness of the adsorption centres.
6.    Tab. 2. 178 ---> 178.0, 180 ---> 180.0; see also 2.24 and 3.37.
7.    Line 335. Atomic charges ---> Atomic charges (CA).
8.    Tab. 4. (i) area  ---> As. (ii) Please, give units for As and charge density index. What about CA? [e]?
9.    Line 346 and Fig. 2, the metallic centre (M1), the surface oxygen (Os). This excuse comes up too late !!!!!
10.    Line 349. OA or Oa??? Check the manuscript.
11.    Figs. 6 and 7. (i) y-axis, AC or CA? (ii) x-axis. The description of this axis is illegible. Please enlarge the font.
12.    Line 370. Elinear? Eadsorbed?

Sincerely,
The reviewer.